REGISTERED REPORT

# Registered report: Kinase-dead BRAF and oncogenic RAS cooperate to drive tumor progression through CRAF

**Ajay Bhargava[1], Madan Anant[1], Hildegard Mack[2], Reproducibility Project: Cancer Biology***

[1]Shakti BioResearch LLC, Woodbridge, United States; [2]University of California, San Francisco, San Francisco, United States

**Abstract** The Reproducibility Project: Cancer Biology seeks to address growing concerns about reproducibility in scientific research by conducting replications of selected experiments from a number of high-profile papers in the field of cancer biology. The papers, which were published between 2010 and 2012, were selected on the basis of citations and Altmetric scores (*Errington et al., 2014*). This Registered Report describes the proposed replication plan of key experiments from "Kinase-dead BRAF and oncogenic RAS cooperate to drive tumor progression through CRAF" by Heidorn and colleagues, published in *Cell* in 2010 (*Heidorn et al., 2010*). The experiments to be replicated are those reported in Figures 1A, 1B, 3A, 3B, and 4D. Heidorn and colleagues report that paradoxical activation of the RAF-RAS-MEK-ERK pathway by BRAF inhibitors when applied to BRAF$^{WT}$ cells is a result of BRAF/CRAF heterodimer formation upon inactivation of BRAF kinase activity, and occurs only in the context of active RAS. The Reproducibility Project: Cancer Biology is a collaboration between the Center for Open Science and Science Exchange, and the results of the replications will be published by *eLife*.

*For correspondence: nicole@ scienceexchange.com

Group author details:
Reproducibility Project: Cancer Biology See page 26

## Introduction

The RAS-RAF-MEK-ERK signaling pathway is routinely disregulated in many forms of cancer. Activating mutations in BRAF are found in almost half of all melanomas, and of these mutations, almost 90% involve a valine to glutamic acid transition at position 600 (BRAF$^{V600E}$) (*Solit and Rosen 2014*). The therapeutic effect of drugs that target this form of BRAF have proved less efficacious than expected, due to an unexpected effect in cells that are BRAF$^{WT}$; in these cells, drugs that target BRAF paradoxically activate rather than repress downstream signaling (*Hall-Jackson et al., 1999a*; *Hall-Jackson et al., 1999b*). In their 2010 paper, Heidorn and colleagues examined the mechanism of action behind this paradoxical activation of MEK/ERK signaling. Heidorn and colleagues first observed that paradoxical activation occurred only in the context of BRAF$^{WT}$ and activated RAS, an observation confirmed by two other groups (*Hatzivassiliou et al., 2010*; *Poulikakos et al., 2010*). Dissecting the mechanism, they reported that the formation of BRAF/CRAF heterodimers was necessary for pathway activation, and formation of those heterodimers required active RAS signaling.

In Figure 1A, Heidorn and colleagues examined pathway activation in response to a range of drugs. The inhibitors, sorafenib, which targets and represses both BRAF and CRAF, PLX4720, which is highly selective for and inhibits the activity of BRAF$^{V600E}$. 885-A, which specifically targets and inhibits BRAF, and the MEK inhibitor PC184352 were examined. As expected, all four drugs blocked MEK/ERK activation in BRAF$^{V600E}$ A375 cells. However, in cells with active RAS, such as D04 cells (BRAF$^{WT}$/NRAS$^{Q61L}$), MEK/ERK signaling was not repressed by PLX4720 or 885-A. This paradoxical activation in BRAF$^{WT}$ cells was also observed by several other groups (*Carnahan et al., 2010*;

*Joseph et al., 2010*; *Lee et al., 2010*; *Kaplan et al., 2011*). This experiment will be replicated in Protocol 1.

Previous work had shown that activated RAS in melanoma signals through CRAF, while normal signaling in healthy melanocytes is accomplished through BRAF (*Dumaz et al., 2006*). To determine if CRAF was required for paradoxical pathway activation, Heidorn and colleagues treated D04 cells with siRNAs targeting NRAS and CRAF. Knockdown of either NRAS or CRAF abrogated activation of MEK/ERK by 885-A, as seen in Figure 1B. This experiment will be replicated in Protocol 2. The necessity of CRAF also explains the lack of activation upon treatment with sorafenib observed in Figure 1A; since sorafenib inhibits both BRAF and CRAF, it does not result in pathway activation.

Since activated RAS is known to drive heterodimerization of BRAF and CRAF (*Weber et al., 2001*), Heidorn and colleagues also tested if drug binding drove heterodimerization of BRAF and CRAF, and if this heterodimerization was dependent on active RAS signaling. In Figure 3A, they transfected D04 cells with a mutant version of CRAF that was unable to bind to RAS (CRAF$^{R89L}$). Immunoprecipitation experiments showed that while CRAF$^{WT}$ was able to bind to BRAF in the presence of activated RAS, CRAF$^{R89L}$ was unable to bind to BRAF. This key experiment will be replicated in Protocol 3.

The authors showed that BRAF binds to CRAF but only in the presence of WT RAS, not oncogenic RAS. In Figure 3B, myc-tagged BRAF or myc-tagged mutant BRAF ($^{R188L}$BRAF) were transfected into D04 cells and treated with either DMSO(-) or 885-A(+). The authors show that mutant of BRAF ($^{R188L}$BRAF) does not bind to CRAF even in the presence of 885-A, which induces RAS activity.

After confirming that drug binding to BRAF drove BRAF binding to CRAF, Heidorn and colleagues tested a kinase dead version of BRAF (BRAF$^{D594A}$) (Figure 4D). Interestingly, this version of BRAF still bound to CRAF, indicating that it is not drug binding per se, but inhibition of BRAF activity, that drives BRAF binding to CRAF and paradoxical activation of MEK/ERK. This key experiment will be replicated in Protocol 4.

Packer and colleagues extended the work of Heidorn and colleagues to examine if other more broadly targeted tyrosine kinase inhibitors were also able to paradoxically activate the RAS-RAF pathway. They observed paradoxical pathway activation in D04 cells after treatment with imatinib, nilotinib, dasatinib, and the BRAF inhibitor SB590885. As in Heidorn et al., paradoxical activation only occurred in cells with BRAF$^{WT}$ and required active RAS, as knockdown of NRAS abrogated the effect. Interestingly, while Heidorn and colleagues reported that knockdown of CRAF alone was able to block paradoxical activation, Packer and colleagues reported that only combined knockdown of BRAF and CRAF was able to block paradoxical activation (*Packer et al., 2011*). Work by Rebocho and colleagues and by Kaplan and colleagues aligned with Heidorn's findings that silencing of CRAF alone was able to abrogate paradoxical activation (*Aplin et al., 2011*; *Rebocho and Marais 2012*). Packer and colleagues also reported that BRAF/CRAF heterodimerization was dependent upon RAS by demonstrating that CRAF$^{R89L}$ was unable to form heterodimers with BRAF (*Packer et al., 2011*).

Activation of NRAS signaling appears to be a key step in acquired drug resistance, supporting the hypothesis that paradoxical activation can only occur in the context of active RAS signaling. Su and colleagues derived a drug-resistant BRAF$^{V600E}$ melanoma cell line by growing A375 cells in the presence of vemurafenib (PLX4032, a BRAF$^{V600E}$ inhibitor). Interestingly, drug resistance was dependent on expression of CRAF, and the resistant lines that emerged had acquired an activating mutation in KRAS (*Su et al., 2012*). Nazarian and colleagues also observed the acquisition of activating mutations in NRAS when they derived PLX4032-resistant cell lines (*Nazarian et al., 2010*). Lidsky and colleagues also showed that increased levels of NRAS were key to vemurafenib resistance, although they did not observe any activating mutations in their resistant cell lines (*Lidsky et al., 2014*).

## Materials and methods

Unless otherwise noted, all protocol information was derived from the original paper, references from the original paper, or information obtained directly from the authors. An asterisk (*) indicates data or information provided by the Reproducibility Project: Cancer Biology core team. A hashtag (#) indicates information provided by the replicating lab.

## Protocol 1: Treatment of BRAF mutant cells with various RAF inhibitors and assessment of activation of ERK

This protocol describes how to treat NRAS mutant D04 cells and NRAS wild-type cells also carrying the $BRAF^{V600E}$ mutation with various BRAF inhibitors and assess ERK phosphorylation by Western blot, as reported in Figure 1A.

### Sampling

- The experiment will be performed independently at least three times for a final power of at least 80%. The original data is qualitative, thus to determine an appropriate number of replicates to initially perform, sample sizes based on a range of potential variance was determined.
  - See Power calculations for details.
- Each experiment consists of eight cohorts:
  - Cohort 1: D04 cells treated with DMSO
  - Cohort 2: D04 cells treated with PD184352
  - Cohort 3: D04 cells treated with sorafenib
  - Cohort 4: D04 cells treated with SB590885
  - Cohort 5: A375 cells treated with DMSO
  - Cohort 6: A375 cells treated with PD184352
  - Cohort 7: A375 cells treated with sorafenib
  - Cohort 8: A375 cells treated with SB590885
- Each cohort will be probed for ppERK and ERK2 by Western blot.

### Materials and reagents

| Reagent | Type | Manufacturer | Catalog # | Comments |
|---|---|---|---|---|
| D04 cells | Cells | Provided by Chris Schmidt, Queensland Institute of Medical Research (QIMR) Berghofer, Australia | | |
| A375 cells | Cells | ATCC | | |
| RPMI | Cell culture media | Life Technologies | 21875-034 | |
| DMEM | Cell culture media | Life Technologies | 41966-029 | |
| FBS | Reagent | Life Technologies | 10270106 | |
| 35-mm culture plates | Material | Corning | CLS430165 | Original not specified |
| Sorafenib | Drug | Selleckchem | S7397 | |
| PD184352 | Drug | Selleckchem | S1020 | |
| SB590885 | Drug | Selleckchem | S2220 | *Replaces Plexxion 885-A |
| DMSO | Reagent | Fisher Scientific | D128-500 | Original not specified |
| PBS | Reagent | Gibco | 10010-023 | Original not specified |
| Tris-HCl | Chemical | Specific brand information will be left up to the discretion of the replicating lab and recorded later | | |
| NaCl | Chemical | | | |
| Igepal | Chemical | | | |
| $Na_3VO_4$ | Chemical | | | |
| NaF | Chemical | | | |
| Leupeptin | Chemical | | | |
| Bradford Assay | Detection Assay | Bio-Rad Laboratories | 5000001 | Original not specified |
| NuPAGE Sample buffer | Buffer | Invitrogen | NP0007 | Original not specified |
| SDS-Page gel (4–12%) | Western blot reagent | Invitrogen | NP0322BOX | Original not specified |
| Nitrocellulose membrane (iBlot) | Western blot reagent | Invitrogen | IB301002 | Original not specified |
| Ponceau stain | Western blot reagent | Sigma-Aldrich | P7170-1L | Original not specified |
| Tris | Chemical | Sigma-Aldrich | T6066 | Original not specified |
| Tween-20 | Chemical | Sigma-Aldrich | P1379 | Original not specified |

*Continued on next page*

*Continued*

| Reagent | Type | Manufacturer | Catalog # | Comments |
|---|---|---|---|---|
| Mouse α-ppERK1/2 | Antibody | Cell Signaling Technology | 9106 | Replaces Sigma M8159 |
| Rabbit α-ERK1/2 | Antibody | Cell Signaling Technology | 9102 | Replaces Santa Cruz Bio sc-154 |
| HRP-conjugated secondary antibody | Western blot reagent | Bio-Rad | 170-5047 | Original not specified |
| ECL Detection Kit | Western blot reagent | Invitrogen | 32132 | Original not specified |

*Suggested as suitable replacement by original authors by personal communication

## Procedure

- All cells will be sent for mycoplasma testing and STR profiling.
- D04 cells are maintained in RPMI supplemented with 10% FBS.
- A375 cells are maintained in DMEM supplemented with 10% FBS.
    - All cell lines are maintained at 37°C with 10% $CO_2$.
- Sorafenib, PD184352, and SB590885 are dissolved in DMSO.

1. Seed 1.0-2 x $10^5$ cells per well of a six-well tissue culture plate (cells should be 80% confluent at the time of drug treatment).
2. Treat cells with drug or equivalent volume vehicle (DMSO, <0.2%) for 4.
    1. 10 µM Sorafenib
    2. 1 µM SB590885
    3. 1 µM PD184352
3. Lyse cells
    1. Place cells on ice and aspirate media.
    2. Wash three times with ice-cold PBS.
    3. Scrape cells into 50–200 µl of Nonidet P40 extraction buffer.
        1. NP40 extraction buffer: 50 mM Tris-HCl, pH 7.5, 150 mM NaCl, 0.55 (v/v) Igepal, 5 mM NaF, 0.2 mM $Na_3VO_4$, 5 µg/ml leupeptin
        2. Incubate on ice for 5min.
    4. Shear cells by passing through a pipette tip several times.
    5. Centrifuge samples at 20,000 x *g* for 5min at 4°C.
    6. Harvest the soluble fraction for further analysis.
    7. [#]Quantify protein concentration using a Bradford assay.
4. Analyze cell lysates by Western blot for phospho-ERK and total ERK.
    1. Load equal amounts of all samples (30–50 µg; approximately half of the lysate) mixed with 4x sample buffer and boiled at 90°C for 5–10min on a [#]4–12% SDS-Page gel.
        1. [#]Run at [#]140v for 55min.
    2. [#]Transfer to a nitrocellulose membrane at 250 mA for 1 hr
        1. *Confirm protein transfer by Ponceau staining and image membrane.
    3. [#]Block membrane in 5% non-fat dried milk in TBST (20 mM Tris pH 7.5, 136 mM NaCl, 0.1% Tween-20).
    4. Incubate membrane at 4°C overnight with antibodies against:
        1. Mouse α-ppERK1/2: 1:1000 dilution
        2. [#]Rabbit α-ERK1/2: 1:1000 dilution
5. [#]Incubate with HRP-conjugated secondary antibody diluted 1:10,000 in 1X TBS for 1 hr at room temperature.
    1. Rinse the membrane twice with TBST.
    2. Wash the membrane twice with TBST for 5 min each.
6. [#]Visualize bands with ECL detection kit according to manufacturer's protocol.
    1. Quantify band intensity.
    2. Normalize pERK to ERK 1/2 for each condition.
7. Repeat independently two additional times.

Deliverables

- Data to be collected:
  - Protein quantification results from Bradford assay.
  - Images of Ponceau stained membranes.
  - Raw images of whole gels with ladders included (as reported in Figure 1A).
  - Densitometric quantification of all bands.

## Confirmatory analysis plan

- Statistical Analysis of the Replication Data:

Note: At the time of analysis, we will perform the Shapiro-Wilk test and generate a quantile-quantile plot to assess the normality of the data. We will also perform Levene's test to assess homoscedasticity. If the data appears skewed, we will perform a transformation in order to proceed with the proposed statistical analysis. If this is not possible, we will perform the equivalent non-parametric test listed.

- Two-way ANOVA on normalized pERK values (to ERK1/2) in A375 or D04 cells treated with PD184352, sorafenib, SB590885, or vehicle (DMSO) with the following planned contrasts with the Bonferroni correction:
  - Normalized pERK band intensity in A375 cells:
    - Vehicle treatment vs. all three drug treatments (PD184352, sorafenib, and SB590885)
  - Normalized pERK band intensity in D04 cells:
    - Vehicle treatment vs. PD184352 and SB590885 treatments
    - Vehicle treatment vs. sorafenib treatment
- Meta-analysis of original and replication attempt effect sizes:
  - The replication data (mean and 95% confidence interval) will be plotted with the original quantified data value displayed as a single point on the same plot for comparison.

## Known differences from the original study

The replication attempt will use D04 and A375 cells and will exclude MM415, MM485, and WM852 cells. It will also exclude the drug PLX4720 and will replace 885-A with its analogue SB590885. The original authors suggest they have found similar results with this analogue (personal communication with Dr. Dhomen). All known differences, if any, are listed in the 'Materials and reagents' section above with the originally used item listed in the comments section. The comments section also lists if the source of original item was not specified. All differences have the same capabilities as the original and are not expected to alter the experimental design.

## Provisions for quality control

All data obtained from the experiment - raw data, data analysis, control data, and quality control data - will be made publicly available, either in the published manuscript or as an open access dataset available on the Open Science Framework (https://osf.io/b1aw6/). Cells will be sent for mycoplasma testing confirming lack of contamination and STR profiling confirming cell line authenticity. The transfer efficiency during the Western blot procedure will be monitored by Ponceau staining.

## Protocol 2: Treatment of NRAS or CRAF silenced D04 cells with SB590885 and assessment of MEK and ERK phosphorylation

This protocol describes treatment of D04 cells transfected with siRNAs targeting NRAS or CRAF with SB590885 and assessment of those cells for activation of MEK and ERK by Western blot, as reported in Figure 1B.

## Sampling

- The experiment will be performed independently at least four times for a final power of at least 80%. The original data is qualitative, thus to determine an appropriate number of replicates to initially perform, sample sizes based on a range of potential variance was determined.

- See Power calculations for details.
- Each experiment consists of six cohorts:
  - Cohort 1: control silenced D04 cells
  - Cohort 2: control silenced D04 cells treated with SB590885
  - Cohort 3: NRAS silenced D04 cells
  - Cohort 4: NRAS silenced D04 cells treated with SB590885
  - Cohort 5: CRAF silenced D04 cells
  - Cohort 6: CRAF silenced D04 cells treated with SB590885
- Each cohort will be probed for NRAS, CRAF, ppMEK, $\alpha$ ppERK, and tubulin by Western blot

## Materials and reagents

| Reagent | Type | Manufacturer | Cat. No. | Comments |
|---|---|---|---|---|
| D04 cells | Cells | Provided by Chris Schmidt, Queensland Institute of Medical Research (QIMR) Berghofer, Australia | | |
| RPMI | Cell culture media | Life Technologies | 21875-034 | |
| FBS | Reagent | Life Technologies | 10270106 | |
| SB590885 | Drug | Selleckchem | S2220 | *Replaces Plexxion 885-A |
| DMSO | Reagent | Fisher Scientific | D128-500 | Original not specified |
| 35 mm tissue culture dishes | Materials | Corning | CLS430165 | Original not specified |
| INTERFERin | Reagent | Polyplus Transfection | 409-01 | |
| CRAF siRNA | siRNA | Synthesis left to the discretion of the replicating lab and will be recorded later | 5'-AAGCACGCTTAGATTG GAATA-3' | |
| NRAS siRNA | siRNA | Synthesis left to the discretion of the replicating lab and will be recorded later | 5'-CATGGCACTGTACTCT TCTCG-3' | |
| Scrambled siRNA | siRNA | Synthesis left to the discretion of the replicating lab and will be recorded later | 5'-AAACCGTC GATTTCACCCGGG-3' | |
| PBS | Reagent | Gibco | 10010-023 | Original not specified |
| Tris-HCl | Chemical | Specific brand information will be left up to the discretion of the replicating lab and recorded later | | |
| NaCl | Chemical | | | |
| Igepal | Chemical | | | |
| $Na_3VO_4$ | Chemical | | | |
| NaF | Chemical | | | |
| Leupeptin | Chemical | | | |
| Bradford Assay | Detection Assay | Bio-Rad Laboratories | 5000001 | Original not specified |
| NuPAGE Sample buffer | Buffer | Invitrogen | NP0007 | Original not specified |
| SDS-Page gel (4–12%) | Western blot reagent | Invitrogen | NP0322BOX | Original not specified |
| Nitrocellulose membrane (iBlot) | Western blot reagent | Invitrogen | IB301002 | Original not specified |
| Ponceau stain | Western blot reagent | Sigma-Aldrich | P7170-1L | Original not specified |
| Tris | Chemical | Sigma-Aldrich | T6066 | Original not specified |
| Tween-20 | Chemical | Sigma-Aldrich | P1379 | Original not specified |
| Mouse $\alpha$ NRAS (C-20) | Antibody | Santa Cruz Biotechnology | sc-159 | |
| Mouse $\alpha$ CRAF | Antibody | BD Transduction Laboratories | 610152 | |
| Rabbit $\alpha$ ppMEK1/2 | Antibody | Cell Signaling Technology | 9121 | |
| Mouse $\alpha$ ppERK1/2 | Antibody | Sigma | M8159 | |
| Mouse $\alpha$ tubulin | Antibody | Sigma | T5168 | |
| HPR-conjugated secondary antibody | Western blot reagent | Bio-Rad | 170-5047 | Original not specified |
| ECL Detection Kit | Western blot reagent | Invitrogen | 32132 | Original not specified |

## Procedure

### Notes

- All cells will be sent for mycoplasma testing and STR profiling.
- D04 cells are maintained in RPMI supplemented with 10% FBS.
  - All cell lines are maintained at 37°C with 10% $CO_2$.
- SB590885 is dissolved in DMSO.

1. Seed 3 x $10^5$ D04 cells per 35-mm plate in 2 ml media.
   1. Let incubate overnight.
2. The next morning, prepare siRNA transfection mixture with INTERFERin according to the manufacturer's protocol, summarized here:
   1. Mix 0.6 μl of 20 μM siRNA with 6 μl INTERERin and 200 μl of serum-free media in RNAse-free tubes.
      1. CRAF siRNA: 5'-AAGCACGCTTAGATTGGAATA-3'
      2. NRAS siRNA: 5'-CATGGCACTGTACTCTTCTCG-3'
      3. Scrambled siRNA control: 5'-AAACCGTC GATTTCACCCGGG-3'
   2. Vortex mixture for 10 s.
   3. Incubate for 5 to 10 min.
   4. Add mixture dropwise to seeded cells in complete media.
   5. Incubate overnight.
3. The next day after transfection, replace with serum free media.
4. 48 hr after siRNA transfection, treat cells with 1 μM SB590885 or equivalent volume vehicle (DMSO, <0.2%) for 4 hr.
5. Lyse cells and harvest extracts as described in Protocol 1 Step 3.
6. Perform Western blots on cell extracts as described in Protocol 1 Step 4.
   a. Blot membranes with the following antibodies:
      1. Rabbit α ppMEK: 1:1000 dilution
      2. Rabbit α ppERK: 1:1000 dilution
      3. Mouse α tubulin: 1:5000 dilution

**Western blot antibody multiplexing**

| Combination | POI | | | Loading control | |
| --- | --- | --- | --- | --- | --- |
| | Description | Working conc. | | Description | Working conc. |
| 1 | Rabit anti-ppMEK (45 kDa) | 1:1000 | | Mouse anti-tubulin (50 kDa) | 1:5000 |
| 2 | Rabbit anti-ppERK (42, 44 kDa) | 1:1000 | | Mouse anti-tubulin (50 kDa) | 1:5000 |

   4. Strip gels with glycine buffer (pH 3.0) containing 1%SDS
   5. Confirm complete stripping and image membranes, block with milk/TBST, and re-probe each gel with one of the following antibodies:
      1. Mouse α NRAS: 1:250 dilution
      2. Mouse α CRAF: 1:1000 dilution
   b. Quantify band intensity.
   c. Normalize NRAS, CRAF, ppMEK, and ppERK to tubulin for each condition.
7. Repeat independently three additional times.

### Deliverables

- Data to be collected:
  - Protein quantification results from Bradford assay.
  - Images of Ponceau stained membranes.
  - Images of whole gels with ladder (as reported in Figure 1B).
  - Densitometric quantification of all bands.

### Confirmatory analysis plan

- Statistical Analysis of the Replication Data:

Note: At the time of analysis, we will calculate Pearson's *r* to check for correlation between the dependent variables, a scatter plot to assess linearity, and a Box's M test to check for equality of covariance matrices. We will also perform the Shapiro-Wilk test and generate a quantile-quantile plot to assess the normality of the data. We will perform Levene's test to assess homoscedasticity. If the data appears skewed, we will perform a transformation in order to proceed with the proposed statistical analysis. If this is not possible, we will perform the equivalent non-parametric test.

- One-way MANOVA comparing the differences between SB590885 treatment and vehicle treatment of normalized band intensities for pMEK and pERK levels in D04 cells transfected with siRNA for NRAS, CRAF, or control with the following Bonferroni-corrected comparisons:
  - Difference in normalized ppMEK levels between SB590885 and vehicle treatment:
    - Control siRNA compared to NRAS siRNA.
    - Control siRNA compared to CRAF siRNA.
  - Difference in normalized ppERK levels between SB590885 and vehicle treatment:
    - Control siRNA compared to NRAS siRNA.
    - Control siRNA compared to CRAF siRNA
- Meta-analysis of original and replication attempt effect sizes:
  - The replication data (mean and 95% confidence interval) will be plotted with the original quantified data value displayed as a single point on the same plot for comparison.

## Known differences from the original study

The replication will replace 885-A with its analogue SB590885. The original authors suggest they have found similar results with this analogue (personal communication with Dr. Dhomen). All known differences, if any, are listed in the 'Materials and reagents' section above with the originally used item listed in the comments section. The comments section also lists if the source of original item was not specified. All differences have the same capabilities as the original and are not expected to alter the experimental design.

## Provisions for quality control

Cells will be sent for mycoplasma testing confirming lack of contamination and STR profiling confirming cell line authenticity. The transfer efficiency during the Western blot procedure will be monitored by Ponceau staining. The membrane will be imaged after stripping to confirm and measure background. All data obtained from the experiment - raw data, data analysis, control data, and quality control data - will be made publicly available, either in the published manuscript or as an open access dataset available on the Open Science Framework (https://osf.io/b1aw6/).

## Protocol 3: Immunoprecipitation of CRAF from SB590885 treated D04 cells expressing myc-tagged CRAF$^{WT}$ or CRAF$^{R89L}$

This protocol describes how to immunoprecipitate myc-tagged CRAF$^{WT}$ or CRAF$^{R89L}$, a mutant form that cannot bind RAS, from D04 cells and probe the pulldown for BRAF, as reported in Figure 3A.

### Sampling

- The experiment will be performed independently at least three times for a final power of at least 80%. The original data is qualitative, thus to determine an appropriate number of replicates to initially perform, sample sizes based on a range of potential variance was determined.
  - See Power calculations for details.
- Each experiment consists of six cohorts:
  - Cohort 1: D04 cells transfected with myc-tagged CRAF$^{WT}$ treated with SB590885
  - Cohort 2: D04 cells transfected with myc-tagged CRAF$^{WT}$ treated with DMSO
  - Cohort 3: D04 cells transfected with myc-tagged CRAF$^{R89L}$ treated with SB590885
  - Cohort 4: D04 cells transfected with myc-tagged CRAF$^{R89L}$ treated with DMSO
  - Cohort 5: D04 cells transfected with empty vector treated with SB590885
  - Cohort 6: D04 cells transfected with empty vector treated with DMSO
- Each cohort will be immunoprecipitated for myc-tagged CRAF and immunoprecipitate and lysates probed for BRAF and myc.

## Materials and reagents

| Reagent | Type | Manufacturer | Catalog # | Comments |
|---|---|---|---|---|
| D04 cells | Cells | Provided by Chris Schmidt, Queensland Institute of Medical Research (QIMR) Berghofer, Australia | | |
| SB590885 | Drug | Selleckchem | S2220 | *Replaces Plexxion 885-A |
| DMSO | Reagent | Fisher Scientific | D128-500 | Original not specified |
| RPMI | Media | Life Technologies | 21875-034 | |
| FBS | Reagent | Life Technologies | 10270106 | |
| Effectene Transfection Reagent | Reagent | Qiagen | 301425 | Replaces Cell Line Nucleofector Kit V (10 RCT) Lonza VACA1003 |
| 35 mm culture dishes | Materials | Corning | CLS430165 | Original not specified |
| Myc-CRAF$^{WT}$ vector | Plasmid | Shared by original authors | | |
| Myc-CRAF$^{R89L}$ vector | Plasmid | Shared by original authors | | |
| Empty vector | Plasmid | Shared by original authors | | |
| PBS | Reagent | Gibco | 10010-023 | Original not specified |
| Tris-HCl | Chemical | Specific brand information will be left up to the discretion of the replicating lab and recorded later | | |
| NaCl | Chemical | | | |
| Igepal | Chemical | | | |
| Na$_3$VO$_4$ | Chemical | | | |
| NaF | Chemical | | | |
| Leupeptin | Chemical | | | |
| Rabbit α myc | Antibody | Abcam | ab9106 | |
| Mouse α BRAF (F-7) | Antibody | Santa Cruz Biotechnology | sc-5284 | |
| Mouse α myc (9B11) (HRP conjugate) | Antibody | Cell Signaling Technology | 2040 | |
| Protein G sepharose beads | Materials | Sigma | P3296 | |
| NuPAGE Sample buffer | Buffer | Invitrogen | NP0007 | Original not specified |
| SDS-Page gel (4–12%) | Western blot reagent | Invitrogen | NP0322BOX | Original not specified |
| Nitrocellulose membrane (iBlot) | Western blot reagent | Invitrogen | IB301002 | Original not specified |
| Ponceau stain | Western blot reagent | Sigma-Aldrich | P7170-1L | Original not specified |
| Tris | Chemical | Sigma-Aldrich | T6066 | Original not specified |
| Tween-20 | Chemical | Sigma-Aldrich | P1379 | Original not specified |
| HPR-conjugated secondary antibody | Western blot reagent | Bio-Rad | 170-5047 | Original not specified |
| ECL Detection Kit | Western blot reagent | Invitrogen | 32132 | Original not specified |

## Procedure

### Notes

- All cells will be sent for mycoplasma testing and STR profiling.
- D04 cells are maintained in RPMI supplemented with 10% FBS.
  - All cell lines are maintained at 37°C with 10% $CO_2$.
- SB590885 is dissolved in DMSO.

1. Transfect D04 cells with vectors containing myc-tagged CRAF$^{wt}$ or CRAF$^{R89L}$.
   1. #Plate 1x10$^6$ cells per well of a six-well plate with 1.6 ml media 1 day before transfection. The cells should be 40–80% confluent on the day of transfection.
   2. #On the day of transfection, dilute 0.4 μg of DNA for each vector in TE buffer, pH 7 with the DNA-condensation buffer, Buffer EC, to a total volume of 100 μl. Add 3.2 μl Enhancer and mix by vortexing.
      1. Empty vector

    2. Myc-CRAF$^{WT}$ vector
    3. Myc-CRAF$^{R89L}$ vector
  3. #Incubate at room temperature for 5 min, centrifuge quickly.
  4. #Add 10 µl Effectene Transfection Reagent to the DNA-Enhancer mixture and mix by pipetting.
  5. #Incubate at room temperature for 10 min.
  6. #Gently aspirate the medium from the plated cells and wash once with 2 ml PBS. Add 1.6 ml fresh medium to the cells.
  7. #Add 600 µl medium to the tube containing transfection complexes and mix by pipetting. Immediately add transfection complexes drop-wise onto plated cells. Gently swirl to mix.
  8. #Incubate for 18 hr after transfection. Replace with fresh medium.
2. 48 hr after transfection, treat cells with 1 µM SB590885 or equivalent volume vehicle (DMSO, <0.2%) for 4 hr.
3. Lyse cells and prepare cell lysate as described in Protocol 1 Step 3.
   1. Save 5–15 µg protein from each lysate to confirm transfection by Western blot below.
4. Immunoprecipitate myc-tagged CRAF proteins
   Note: 2-3 35 mm wells of protein lysed in 300 µl NP40 buffer are needed for IP reaction.
   1. Immunoprecipitate the Myc-tagged proteins by adding 2 µg rabbit anti-myc antibody and incubate overnight at 4°C.
   2. Capture the antibody-protein complex by adding 20 µl of a 1:1 Protein G sepharose 4B beads mixture in NP40 extraction buffer.
      1. NP40 extraction buffer: 50 mM Tris-HCl, pH 7.5, 150 mM NaCl, 0.55 (v/v) Igepal, 5 mM NaF, 0.2 mM $Na_3VO_4$, 5 µg/ml leupeptin.
      2. Incubate on ice for 5 min.
      3. Mix on a rotating wheel for 2 hr at 4°C.
   3. Wash IPs three times with 300 µl NP40 extraction buffer.
   4. Elute protein complex from beads with NuPage sample buffer
5. Run IPs and lysate on an SDS-PAGE gel as described in Protocol 1 Step 4.
   1. Probe with the following antibodies:
      1. Mouse $\alpha$ BRAF: 1:2000 dilution
      2. Mouse $\alpha$ myc: 1:1000 dilution
   2. Quantify band intensity.
   3. Normalize IP $\alpha$ BRAF to IP $\alpha$ myc-CRAF for each condition from IP band intensities.
6. Repeat independently two additional times.

## Deliverables

- Data to be collected:
  - Protein quantification results from Bradford assay.
  - Images of Ponceau stained membranes.
  - Transfection QC images of whole gels with ladder.
  - Images of whole gels with ladder (as reported in Figure 3A).
  - Densitometric quantification of all bands.

## Confirmatory analysis plan

- Statistical Analysis of the Replication Data:

Note: At the time of analysis, we will perform the Shapiro-Wilk test and generate a quantile-quantile plot to assess the normality of the data. We will also perform Levene's test to assess homoscedasticity. If the data appears skewed, we will perform a transformation in order to proceed with the proposed statistical analysis. If this is not possible, we will perform the equivalent non-parametric test listed.

- Two-way ANOVA comparing normalized IP BRAF (to IP $\alpha$ myc) band intensity in D04 cells transfected with Myc-CRAF$^{WT}$ vector or Myc-CRAF$^{R89L}$ vector with or without SB590885 drug treatment, and the following Bonferroni-corrected comparisons:
  - Normalized IP BRAF band intensity in cells with Myc-CRAF$^{WT}$ vector with SB590885 treatment vs. vehicle treatment.

- Normalized IP BRAF band intensity in cells with Myc- CRAF$^{R89L}$ vector with SB590885 treatment vs. vehicle treatment.
  - Meta-analysis of original and replication attempt effect sizes:
    - The replication data (mean and 95% confidence interval) will be plotted with the original quantified data value displayed as a single point on the same plot for comparison.

## Known differences from the original study

The transfection method using Nucleofectin Solution V and electroporation will be replaced with a lipid-based method using Effectene Transfection Reagent, and protocol will be changed according to Manufacturer's instructions. This difference in transfection protocol might lead to differences in expression that could lead to differences in results. The replication will replace 885-A with its analogue SB590885. The original authors suggest they have found similar results with this analogue (personal communication with Dr. Dhomen). All known differences, if any, are listed in the 'Materials and reagents' section above with the originally used item listed in the comments section. The comments section also lists if the source of original item was not specified. All differences have the same capabilities as the original and are not expected to alter the experimental design.

## Provisions for quality control

Cells will be sent for mycoplasma testing confirming lack of contamination and STR profiling confirming cell line authenticity. Transfection will be confirmed with Western blot. The transfer efficiency during the Western blot procedure will be monitored by Ponceau staining. All data obtained from the experiment - raw data, data analysis, control data, and quality control data - will be made publicly available, either as a published manuscript or as an open access dataset available on the Open Science Framework (https://osf.io/b1aw6/).

## Protocol 4: Immunoprecipitation of BRAF from SB590885 treated D04 cells expressing myc-tagged BRAF$^{WT}$ or BRAF$^{R188L}$

This protocol describes how to immunoprecipitate myc-tagged BRAF$^{WT}$ or BRAF$^{R188L}$, a mutant form that cannot bind RAS, from D04 cells and probe the pulldown for CRAF, as reported in Figure 3B.

### Sampling

- The experiment will be performed independently at least three times for a final power of at least 80%. The original data is qualitative, thus to determine an appropriate number of replicates to initially perform, sample sizes based on a range of potential variance was determined.
  - See Power calculations for details.
- Each experiment consists of six cohorts:
  - Cohort 1: D04 cells transfected with myc-tagged BRAF$^{WT}$ treated with SB590885
  - Cohort 2: D04 cells transfected with myc-tagged BRAF$^{WT}$ treated with DMSO
  - Cohort 3: D04 cells transfected with myc-tagged BRAF$^{R188L}$ treated with SB590885
  - Cohort 4: D04 cells transfected with myc-tagged BRAF$^{R188L}$ treated with DMSO
  - Cohort 5: D04 cells transfected with empty vector treated with SB590885
  - Cohort 6: D04 cells transfected with empty vector treated with DMSO
- Each cohort will be immunoprecipitated for myc-tagged BRAF and immunoprecipitate and lysates probed for CRAF and myc.

### Materials and reagents

| Reagent | Type | Manufacturer | Catalog # | Comments |
|---|---|---|---|---|
| D04 cells | Cells | Provided by Chris Schmidt, Queensland Institute of Medical Research (QIMR) Berghofer, Australia | | |
| SB590885 | Drug | Selleckchem | S2220 | *Replaces Plexxion 885-A |

*Continued on next page*

*Continued*

| Reagent | Type | Manufacturer | Catalog # | Comments |
|---------|------|--------------|-----------|----------|
| DMSO | Reagent | Fisher Scientific | D128-500 | Original not specified |
| RPMI | Media | Life Technologies | 21875-034 | |
| FBS | Reagent | Life Technologies | 10270106 | |
| Effectene Transfection Reagent | Reagent | Qiagen | 301425 | Replaces Cell Line Nucleofector Kit V (10 RCT) Lonza VACA1003 |
| 35-mm culture dishes | Materials | Corning | CLS430165 | Original not specified |
| Myc-BRAF$^{WT}$ vector | Plasmid | Shared by original authors | | |
| Myc-BRAF$^{R188L}$ vector | Plasmid | Shared by original authors | | |
| Empty vector | Plasmid | Shared by original authors | | |
| PBS | Reagent | Gibco | 10010-023 | Original not specified |
| Tris-HCl | Chemical | Specific brand information will be left up to the discretion of the replicating lab and recorded later | | |
| NaCl | Chemical | | | |
| Igepal | Chemical | | | |
| Na$_3$VO$_4$ | Chemical | | | |
| NaF | Chemical | | | |
| Leupeptin | Chemical | | | |
| Rabbit anti-myc | Antibody | Abcam | ab9106 | |
| mouse anti-CRAF | Antibody | BD Transduction Laboratories | 610152 | |
| Mouse α myc (9B11) (HRP conjugate) | Antibody | Cell Signaling Technology | 2040 | |
| Protein G sepharose beads | Materials | Sigma | P3296 | |
| NuPAGE Sample buffer | Buffer | Invitrogen | NP0007 | Original not specified |
| SDS-Page gel (4–12%) | Western blot reagent | Invitrogen | NP0322BOX | Original not specified |
| Nitrocellulose membrane (iBlot) | Western blot reagent | Invitrogen | IB301002 | Original not specified |
| Ponceau stain | Western blot reagent | Sigma-Aldrich | P7170-1L | Original not specified |
| Tris | Chemical | Sigma-Aldrich | T6066 | Original not specified |
| Tween-20 | Chemical | Sigma-Aldrich | P1379 | Original not specified |
| HPR-conjugated secondary antibody | Western blot reagent | Bio-Rad | 170-5047 | Original not specified |
| ECL Detection Kit | Western blot reagent | Invitrogen | 32132 | Original not specified |

## Procedure

Notes:

- All cells will be sent for mycoplasma testing and STR profiling.
- D04 cells are maintained in RPMI supplemented with 10% FBS.
    - All cell lines are maintained at 37°C with 10% $CO_2$.
- SB590885 is dissolved in DMSO.

1. Transfect D04 cells with vectors containing myc-tagged BRAF$^{wt}$ or BRAF$^{R188L}$ as described in Protocol 3 Step 1.
2. 48 hr after transfection, treat cells with 1 μM SB590885 or equivalent volume vehicle (DMSO, <0.2%) for 4 hr.
3. Lyse cells and prepare cell lysate as described in Protocol 1 Step 3.
    1. Save 5-15 μg protein from each lysate to confirm transfection by Western blot below.
4. Immunoprecipitate myc-tagged CRAF proteins
   Note: 2-3 35 mm wells of protein lysed in 300 μl NP40 buffer are needed for IP reaction.
    1. Immunoprecipitate the Myc-tagged proteins by adding 2 μg rabbit anti-myc antibody and incubate overnight at 4°C.

 2. Capture the antibody-protein complex by adding 20 µl of a 1:1 Protein G sepharose 4B beads mixture in NP40 extraction buffer.
 1. NP40 extraction buffer: 50 mM Tris-HCl, pH 7.5, 150 mM NaCl, 0.55 (v/v) Igepal, 5 mM NaF, 0.2 mM $Na_3VO_4$, 5 µg/ml leupeptin.
 2. Incubate on ice for 5 min.
 3. Mix on a rotating wheel for 2 hr at 4°C.
 3. Wash IPs three times with 300 µl NP40 extraction buffer.
 4. Elute protein complex from beads with NuPage sample buffer
 5. Run IPs and lysate on an SDS-PAGE gel as described in Protocol 1 Step 4.
 1. Probe with the following antibodies:
 1. Mouse $\alpha$ CRAF: 1:1000 dilution
 2. Mouse $\alpha$ myc: 1:1000 dilution
 2. Quantify band intensity.
 3. Normalize IP $\alpha$ CRAF to IP $\alpha$ myc-BRAF for each condition from IP band intensities.
 6. Repeat independently two additional times.

## Deliverables

- Data to be collected:
  - Protein quantification results from Bradford assay.
  - Images of Ponceau stained membranes.
  - Transfection QC images of whole gels with ladder.
  - Images of whole gels with ladder (as reported in Figure 3A).
  - Densitometric quantification of all bands.

## Confirmatory analysis plan

- Statistical Analysis of the Replication Data:

Note: At the time of analysis, we will perform the Shapiro-Wilk test and generate a quantile-quantile plot to assess the normality of the data. We will also perform Levene's test to assess homoscedasticity. If the data appears skewed, we will perform a transformation in order to proceed with the proposed statistical analysis. If this is not possible, we will perform the equivalent non-parametric test listed.

- Two-way ANOVA comparing normalized IP CRAF (to IP $\alpha$ myc) band intensity in D04 cells transfected with Myc-BRAF$^{WT}$ vector or Myc-BRAF$^{R188L}$ vector with or without SB590885 drug treatment, and the following Bonferroni-corrected comparisons:
  - Normalized IP CRAF band intensity in cells with Myc-BRAF$^{WT}$ vector with SB590885 treatment vs. vehicle treatment.
  - Normalized IP CRAF band intensity in cells with Myc- BRAF$^{R188L}$ vector with SB590885 treatment vs. vehicle treatment.
- Meta-analysis of original and replication attempt effect sizes:
  - The replication data (mean and 95% confidence interval) will be plotted with the original quantified data value displayed as a single point on the same plot for comparison.

## Known differences from the original study

The transfection method using Nucleofectin Solution V and electroporation will be replaced with a lipid-based method using Effectene Transfection Reagent, and protocol will be changed according to Manufacturer's instructions. The replication will replace 885-A with its analogue SB590885. The original authors suggest they have found similar results with this analogue (personal communication with Dr. Dhomen). All known differences, if any, are listed in the 'Materials and reagents' section above with the originally used item listed in the comments section. The comments section also lists if the source of original item was not specified. All differences have the same capabilities as the original and are not expected to alter the experimental design.

## Provisions for quality control

Cells will be sent for mycoplasma testing confirming lack of contamination and STR profiling confirming cell line authenticity. Transfection will be confirmed with Western blot. The transfer efficiency during the Western blot procedure will be monitored by Ponceau staining. All data obtained from the experiment - raw data, data analysis, control data, and quality control data - will be made publicly available, either as a published manuscript or as an open access dataset available on the Open Science Framework (https://osf.io/b1aw6/).

## Protocol 5: Expression of BRAF kinase dead mutant in D04 cells and its effect on CRAF binding

This protocol describes how to transiently express myc-tagged BRAF$^{WT}$ or BRAF$^{D59A}$ in D04 cells and assess CRAF binding by immunoprecipitation and blotting, as reported in Figure 4D.

### Sampling

- The experiment will be performed independently at least three times for a minimum power of 80%. The original data is qualitative, thus to determine an appropriate number of replicates to initially perform, sample sizes based on a range of potential variance was determined.
    - See Power Calculations for details.
- Each experiment consists of three cohorts:
    - Cohort 1: D04 cells transfected with myc-tagged BRAF$^{WT}$
    - Cohort 2: D04 cells transfected with myc-tagged BRAF$^{D594A}$
    - Cohort 3: D04 cells transfected with empty vector
    - Untreated cells are immunoprecipitated with $\alpha$ myc and levels of myc-BRAF and CRAF are assessed by immunoblotting.

### Materials and reagents

| Reagent | Type | Manufacturer | Catalog # | Comments |
|---|---|---|---|---|
| D04 cells | Cells | Provided by Chris Schmidt, Queensland Institute of Medical Research (QIMR) Berghofer, Australia | | |
| RPMI | Media | Life Technologies | 21875-034 | |
| FBS | Reagent | Life Technologies | 10270106 | |
| Effectene Transfection Reagent | Reagent | Qiagen | 301425 | Replaces Cell Line Nucleofector Kit V (10 RCT) Lonza VACA1003 |
| Myc-BRAF$^{WT}$ vector | Plasmid | Shared by original author | | |
| Myc-BRAF$^{D594A}$ vector | Plasmid | Shared by original author | | |
| Empty vector | Plasmid | Shared by original author | | |
| 35 mm culture dishes | Materials | | | |
| PBS | Reagent | Gibco | 10010-023 | Original not specified |
| Tris-HCl | Chemical | Specific brand information will be left up to the discretion of the replicating lab and recorded later | | |
| NaCl | Chemical | | | |
| Igepal | Chemical | | | |
| Na$_3$VO$_4$ | Chemical | | | |
| NaF | Chemical | | | |
| Leupeptin | Chemical | | | |
| Rabbit $\alpha$ myc | Antibody | Abcam | ab9106 | |
| Mouse $\alpha$ CRAF (for Western blotting) | Antibody | BD Transduction Laboratories | 610152 | |
| Mouse $\alpha$ myc (9B11) (HRP conjugate) | Antibody | Cell Signaling Technology | 2040 | |
| Protein G sepharose beads | Materials | Sigma | P3296 | |
| NuPAGE Sample buffer | Buffer | Invitrogen | NP0007 | Original not specified |

*Continued on next page*

*Continued*

| Reagent | Type | Manufacturer | Catalog # | Comments |
|---------|------|--------------|-----------|----------|
| SDS-Page gel (4–12%) | Western blot reagent | Invitrogen | NP0322BOX | Original not specified |
| Nitrocellulose membrane (iBlot) | Western blot reagent | Invitrogen | IB301002 | Original not specified |
| Ponceau stain | Western blot reagent | Sigma-Aldrich | P7170-1L | Original not specified |
| Tris | Chemical | Sigma-Aldrich | T6066 | Original not specified |
| Tween-20 | Chemical | Sigma-Aldrich | P1379 | Original not specified |
| HPR-conjugated secondary antibody | Western blot reagent | Bio-Rad | 170-5047 | Original not specified |
| ECL Detection Kit | Western blot reagent | Invitrogen | 32132 | Original not specified |

## Procedure

Notes:

- All cells will be sent for mycoplasma testing and STR profiling.
- D04 cells are maintained in RPMI supplemented with 10% FBS.
  - All cell lines are maintained at 37°C with 10% $CO_2$.

1. Transiently transfect D04 cells with the following vectors as described in Protocol 3 step 1.
   1. Myc-BRAF$^{WT}$ vector
   2. Myc-BRAF$^{D594A}$ vector
   3. Empty vector
2. Lyse cells and prepare cell lysates as described in Protocol 1 Step 3.
   1. Save 5–15 $\mu$g protein from each lysate to confirm transfection by Western blot below.
3. Immunoprecipitate myc-tagged BRAF proteins as described in Protocol 3 Step 4.
4. Run IPs and lysate on SDS-PAGE gel as described in Protocol 1 Step 4.
   1. Probe with the following antibodies:
      1. Mouse $\alpha$ CRAF: 1:1000 dilution
      2. Mouse $\alpha$ myc: 1:1000 dilution
   2. Quantify band intensity.
   3. Normalize IP $\alpha$ CRAF to IP $\alpha$ myc-BRAF for each condition from IP band intensities.
5. Repeat independently two additional times.

## Deliverables

- Data to be collected:
  - Protein quantification results from Bradford assay.
  - Images of Ponceau stained membranes.
  - Images of whole gels (as reported in Figure 4D).
  - Densitometric quantification of all bands.
  - Any data pertaining to cell growth conditions optimization, if performed.

## Confirmatory analysis plan

- Statistical Analysis of the Replication Data:
  - A two sample Welch's *t*-test comparing normalized IP CRAF (using IP myc-BRAF band intensity) in D04 cells transfected with Myc-BRAF$^{WT}$ vector vs. Myc-BRAF$^{D594A}$ vector
- Meta-analysis of original and replication attempt effect sizes:
  - The replication data (mean and 95% confidence interval) will be plotted with the original quantified data value displayed as a single point on the same plot for comparison.

## Known differences from the original study

All known differences, if any, are listed in the 'Materials and reagents' section above with the originally used item listed in the comments section. The comments section also lists if the source of

original item was not specified. All differences have the same capabilities as the original and are not expected to alter the experimental design.

## Provisions for quality control

Cells will be sent for mycoplasma testing confirming lack of contamination and STR profiling confirming cell line authenticity. Transfection will be confirmed with Western blot. The transfer efficiency during the Western blot procedure will be monitored by Ponceau staining. All data obtained from the experiment - raw data, data analysis, control data, and quality control data - will be made publicly available, either as a published manuscript or as an open access dataset available on the Open Science Framework (https://osf.io/b1aw6/). Cells will be sent for mycoplasma testing confirming lack of contamination and STR profiling confirming cell line authenticity.

## Power calculations

For additional details on power calculations, please see analysis scripts and associated files on the Open Science Framework:

https://osf.io/eaktg/

## **Protocol 1**

### Summary of original data

- The original data presented is qualitative (images of Western blots). We used Image Studio Lite (LICOR) to perform densitometric analysis of the presented bands. We then performed a priori power calculations with a range of assumed standard deviations to determine the number of replicates to perform.
- Note: band intensity quantified from the image reported in Figure 1A:

| Cell type | Drug | Band intensity normalized total ERK | Assumed N |
|-----------|------|-------------------------------------|-----------|
| A375 | Control | 1.3864 | 3 |
| | PD | 0.0127 | 3 |
| | SF | 0.0257 | 3 |
| | 885-A | 0.0510 | 3 |
| D04 | Control | 0.1315 | 3 |
| | PD | 0.0198 | 3 |
| | SF | 0.0123 | 3 |
| | 885-A | 0.6650 | 3 |

- The original data does not indicate the error associated with multiple biological replicates. To identify a suitable sample size, power calculations were performed using different levels of relative variance.

### Test family

- Two-way ANOVA (2 x 4) fixed effects, special, main effects and interactions; alpha error = 0.05 followed by Bonferroni corrected comparisons

### Power calculations

- Power calculations were performed using R software version 3.2.1 (*R Core Team, 2014*) and G*Power (version 3.1.7) (*Faul et al., 2007*)

| Groups | Estimated variance | F test statistic $F_{(3,16)}$ interaction | Partial $\eta^2$ | Effect size $f$ | A priori power | Total sample size (8 groups) |
|---|---|---|---|---|---|---|
| A375 or D04 cells treated with drugs or control | 2% | 7743.50 | 0.9993 | 38.112 | 99.9% | 9 |
| | 15% | 137.662 | 0.9627 | 5.080 | 98.8% | 10 |
| | 28% | 39.507 | 0.8811 | 2.722 | 96.0% | 11 |
| | 40% | 19.359 | 0.7840 | 1.905 | 91.6% | 12 |

## Test family

- *F* test, ANOVA: Fixed effects, special, main effects and interactions, Bonferroni's correction: alpha error = 0.01667

Power Calculations performed with G*Power software, version 3.1.7 (*Faul et al., 2007*).

ANOVA F test statistic and partial $\eta^2$ performed with R software, version 3.2.1 (*Team, 2015*). Partial $\eta^2$ calculated from (*Lakens, 2013*).

For A375 cells, comparisons are between DMSO and all Drug Treatments (PD184352, sorafenib, and 885-A)

| Groups | Cell line | Variance estimate | F test statistic $Fc_{(1,16)}$ | Partial $\eta^2$ | Effect size $f$ | A priori power | Total sample size (8 groups) |
|---|---|---|---|---|---|---|---|
| DMSO vs all Drug Treatments | A375 | 2% | 34711.2 | 0.9995 | 46.58 | 99.9% | 9 |
| | A375 | 15% | 617.09 | 0.9747 | 6.210 | 99.8% | 10 |
| | A375 | 28% | 177.10 | 0.9171 | 3.327 | 84.2% | 10 |
| | A375 | 40% | 86.78 | 0.8443 | 2.329 | 92.7% | 11 |

For D04 cells, comparisons are between DMSO and PD184352 and sorafenib, and between DMSO and 885-A

| Groups | Cell line | Variance estimate | F test statistic $Fc_{(1,16)}$ | Partial $\eta^2$ | Effect size $f$ | A priori power | Total sample size (8 groups) |
|---|---|---|---|---|---|---|---|
| DMSO vs. PD184352 and sorafenib | D04 | 2% | 223.55 | 0.9332 | 3.7379 | 90.2% | 10 |
| | D04 | 15% | 3.9741 | 0.1990 | 0.4984 | 80.4% | 46 |
| | D04 | 28% | 1.1405 | 0.0665 | 0.2670 | 80.0% | 150 |
| | D04 | 40% | 0.5589 | 0.0337 | 0.1869 | 80.0% | 303 |
| DMSO vs. 885A | D04 | 2% | 3580.31 | 0.9955 | 14.959 | 99.9% | 10 |
| | D04 | 15% | 63.6498 | 0.7991 | 1.9945 | 84.0% | 11 |
| | D04 | 28% | 18.2668 | 0.5331 | 1.0685 | 80.8% | 15 |
| | D04 | 40% | 8.9507 | 0.3587 | 0.7479 | 80.1% | 23 |

- Based on these power calculations, we will run the experiment three times. Each time, we will quantify band intensity. We will determine the standard deviation of band intensity across the biological replicates and combine this with the effect size from the original study to calculate the number of replicates necessary to reach a power of at least 80%. We will then perform additional replicates, if required, to ensure the experiment has more than 80% power to detect the original effect.

## Protocol 2
### Summary of original data

- The original data presented is qualitative (images of Western blots). We used Image Studio Lite (LICOR) to perform densitometric analysis of the presented bands. We then performed a priori power calculations with a range of assumed standard deviations to determine the number of replicates to perform.
- Note: band intensity quantified from the image reported in Figure 1B:

| Target | siRNA | Band intensity normalized to tubulin for transfected cells treated with 885-A minus DMSO | Assumed N |
|---|---|---|---|
| pMEK | Control | 0.836493931 | 3 |
| | NRAS | 0.0695447 | 3 |
| | CRAF | 0.3538748 | 3 |
| pERK | Control | 0.8769868 | 3 |
| | NRAS | 0.498252598 | 3 |
| | CRAF | 0.653649416 | 3 |

### Test family

- Due to the lack of raw original data, we are unable to perform power calculations using a MANOVA. We are determining sample size using two one-way ANOVAs.
- Two, one-way ANOVAs (Bonferroni corrected) on the difference in the normalized band intensity for pMEK and pERK separately in transfected cells treated with 885-A minus DMSO followed by Bonferroni corrected comparisons for the following groups:
  - pMEK and pERK each:
    - Compare the difference in band intensity in cells transfected with control siRNA and treated with 885-A minus control siRNA with DMSO (Control siRNA Difference) vs. the difference in band intensity in cells transfected with NRAS siRNA and treated with 885-A minus NRAS siRNA with DMSO (NRAS siRNA Difference)
    - Compare the difference in band intensity in cells transfected with control siRNA and treated with 885-A minus control siRNA with DMSO (Control siRNA Difference) vs. the difference in band intensity in cells transfected with CRAF siRNA and treated with 885-A minus CRAF siRNA with DMSO (CRAF siRNA Difference)

### Power calculations

- Power calculations were performed using R software version 3.1.2 (*R Core Team, 2014*) and G*Power (version 3.1.7) (*Faul et al., 2007*)

### pMEK

- 2% variance:
  - ANOVA: Fixed effects, omnibus, one-way, Bonferroni corrected alpha error = 0.025

| Groups | F test statistic | Partial $\eta^2$ | Effect size f | A priori power | Total sample size |
|---|---|---|---|---|---|
| D04 cells silenced for NRAS or CRAF and exposed to Drug Treatment | $F_{(2,6)}$ = 1019.1 | 0.9971 | 18.5426 | >99.9% | 6 (3 groups) |

- Bonferroni- corrected planned comparisons; alpha error = 0.0125

| Group 1 | Group 2 | Effect size d | A priori power | Sample size per group |
|---|---|---|---|---|
| Control siRNA Difference | NRAS siRNA Difference | 36.4575 | 99.3%[1] | 2[1] |
| Control siRNA Difference | CRAF siRNA Difference | 8.6916 | 99.9% | 3 |

- 15% variance:
  - ANOVA: Fixed effects, omnibus, one-way, Bonferroni corrected alpha error = 0.025

| Groups | F test statistic | Partial $\eta^2$ | Effect size f | A priori power | Total sample size |
|---|---|---|---|---|---|
| D04 cells silenced for NRAS or CRAF and exposed to Drug Treatment | $F_{(2,6)} = 72.467$ | 0.9602 | 4.9118 | 99.5% | 6 (3 groups) |

- Bonferroni- corrected planned comparisons; alpha error = 0.0125

| Group 1 | Group 2 | Effect size d | A priori power | Sample size per group |
|---|---|---|---|---|
| Control siRNA Difference | NRAS siRNA Difference | 9.7218 | >99.9% | 3 |
| Control siRNA Difference | CRAF siRNA Difference | 2.3177 | 80.9% | 6 |

- 28% variance:
  - ANOVA: Fixed effects, omnibus, one-way, Bonferroni corrected alpha error = 0.025

| Groups | F test statistic | Partial $\eta^2$ | Effect size f | A priori power | Total sample size |
|---|---|---|---|---|---|
| D04 cells silenced for NRAS or CRAF and exposed to Drug Treatment | $F_{(2,6)} = 20.797$ | 0.8739 | 2.6325 | 99.8% | 9 (3 groups) |

- Bonferroni- corrected planned comparisons; alpha error = 0.0125

| Group 1 | Group 2 | Effect size d | A priori power | Sample size per group |
|---|---|---|---|---|
| Control siRNA Difference | NRAS siRNA Difference | 5.2081 | 89.9% | 3 |
| Control siRNA Difference | CRAF siRNA Difference | 1.2416 | 82.7% | 17 |

- 40% variance:
  - ANOVA: Fixed effects, omnibus, one-way, Bonferroni corrected alpha error = 0.025

| Groups | F test statistic | Partial $\eta^2$ | Effect size f | A priori power | Total sample size |
|---|---|---|---|---|---|
| D04 cells silenced for NRAS or CRAF and exposed to Drug Treatment | $F_{(2,6)} = 10.191$ | 0.7726 | 1.8432 | 90.8% | 9 (3 groups) |

- Bonferroni- corrected planned comparisons; alpha error = 0.0125

| Group 1 | Group 2 | Effect size d | A priori power | Sample size per group |
|---|---|---|---|---|
| Control siRNA Difference | NRAS siRNA Difference | 3.6457 | 89.7% | 4 |
| Control siRNA Difference | CRAF siRNA Difference | 0.8692 | 81.4% | 32 |

## pERK

- 2% variance:
  - ANOVA: Fixed effects, omnibus, one-way, Bonferroni corrected alpha error = 0.025

| Groups | F test statistic | Partial $\eta^2$ | Effect size f | A priori power | Total sample size |
|---|---|---|---|---|---|
| D04 cells silenced for NRAS or CRAF and exposed to Drug Treatment | $F_{(2,6)}$ = 141.13 | 0.9792 | 6.8613 | >99.9% | 6 (3 groups) |

- Bonferroni- corrected planned comparisons; alpha error = 0.0125

| Group 1 | Group 2 | Effect size d | A priori power | Sample size per group |
|---|---|---|---|---|
| Control siRNA Difference | NRAS siRNA Difference | 13.6467 | 90.2% | 2 |
| Control siRNA Difference | CRAF siRNA Difference | 8.0474 | 99.9% | 3 |

- 15% variance:
  - ANOVA: Fixed effects, omnibus, one-way, Bonferroni corrected alpha error = 0.025

| Groups | F test statistic | Partial $\eta^2$ | Effect size f | A priori power | Total sample size |
|---|---|---|---|---|---|
| D04 cells silenced for NRAS or CRAF and exposed to Drug Treatment | $F_{(2,6)}$ = 10.036 | 0.7699 | 1.8292 | 90.3% | 9 (3 groups) |

- Bonferroni- corrected planned comparisons; alpha error = 0.0125

| Group 1 | Group 2 | Effect size d | A priori power | Sample size per group |
|---|---|---|---|---|
| Control siRNA Difference | NRAS siRNA Difference | 3.6391 | 89.3% | 4 |
| Control siRNA Difference | CRAF siRNA Difference | 2.1460 | 83.7% | 7 |

- 28% variance:
  - ANOVA: Fixed effects, omnibus, one-way, Bonferroni corrected alpha error = 0.025

| Groups | F test statistic | Partial $\eta^2$ | Effect size f | A priori power | Total sample size |
|---|---|---|---|---|---|
| D04 cells silenced for NRAS or CRAF and exposed to Drug Treatment | $F_{(2,6)}$ = 2.8802 | 0.4898 | 0.9798 | 86.4% | 18 (3 groups) |

- Bonferroni- corrected planned comparisons; alpha error = 0.0125

| Group 1 | Group 2 | Effect size d | A priori power | Sample size per group |
|---------|---------|---------------|----------------|------------------------|
| Control siRNA Difference | NRAS siRNA Difference | 1.9495 | 83.1% | 8 |
| Control siRNA Difference | CRAF siRNA Difference | 1.1496 | 81.4% | 19 |

- 40% variance:
  - ANOVA: Fixed effects, omnibus, one-way, Bonferroni corrected alpha error = 0.025

| Groups | F test statistic | Partial $\eta^2$ | Effect size f | A priori power | Total sample size |
|--------|------------------|-------------------|----------------|----------------|--------------------|
| D04 cells silenced for NRAS or CRAF and exposed to Drug Treatment | $F_{(2,6)} = 1.4113$ | 0.3199 | 0.6858 | 82.9% | 30 (3 groups) |

- Bonferroni- corrected planned comparisons; alpha error = 0.0125

| Group 1 | Group 2 | Effect size d | A priori power | Sample size per group |
|---------|---------|---------------|----------------|------------------------|
| Control siRNA Difference | NRAS siRNA Difference | 1.3647 | 81.4% | 14 |
| Control siRNA Difference | CRAF siRNA Difference | 0.8047 | 81.3% | 37 |

- Based on these power calculations, we will run the experiment four times. Each time, we will quantify band intensity. We will determine the standard deviation of band intensity across the biological replicates and combine this with the effect size from the original study to calculate the number of replicates necessary to reach a power of at least 80%. We will then perform additional replicates, if required, to ensure the experiment has more than 80% power to detect the original effect.

## Protocol 3
## Summary of original data

- The original data presented is qualitative (images of Western blots). We used Image Studio Lite (LICOR) to perform densitometric analysis of the presented bands. We then performed a priori power calculations with a range of assumed standard deviations to determine the number of replicates to perform.
- Note: band intensity quantified from the image reported in Figure 3A:

| Target | Myc-eptitope tagged vector | Drug | Band intensity normalized to IP myc | Assumed N |
|--------|----------------------------|------|--------------------------------------|-----------|
| BRAF | CRAF | 885-A | 0.01904 | 3 |
| | | DMSO | 0.94756 | 3 |
| | R89L | 885-A | 0.27776 | 3 |
| | | DMSO | 0.65427 | 3 |

## Test family

- Two-way ANOVA (2 x 2) on BRAF values followed by Bonferroni corrected comparisons for the following groups:
  - Compare the band intensity of BRAF in myc-tagged CRAF[WT] or CRAF[R89L] in cells treated with or without 885-A

## Power calculations

- Power calculations were performed using R software version 3.1.2 (*R Core Team, 2014*) and G*Power (version 3.1.7) (*Faul et al., 2007*)
- 2% variance:
  - ANOVA: Fixed effects, special, main effects, and interactions; alpha error = 0.05

| Groups | F test statistic | Partial eta$^2$ | Effect size $f$ | A priori power | Total sample size |
|---|---|---|---|---|---|
| myc-tagged CRAF$^{WT}$ or CRAF$^{R89L}$in cells with or without 885-A | $F_{(1.8)} =$ 1628.39 (interaction) | 0.9951 | 14.267 | 98.7% | 5 (4 groups) |

- Bonferroni- corrected planned comparisons; alpha error = 0.025

| Group 1 | Group 2 | Effect size d | A priori power | Sample size per group |
|---|---|---|---|---|
| CRAF +885A | CRAF +DMSO | 69.2756 | 99.9% | 2 |
| R89L +885A | R89L +DMSO | 37.4562 | 99.9% | 2 |

- 15% variance:
  - ANOVA: Fixed effects, special, main effects, and interactions; alpha error = 0.05

| Groups | F test statistic | Partial eta$^2$ | Effect size $f$ | A priori power | Total sample size |
|---|---|---|---|---|---|
| myc-tagged CRAF$^{WT}$ or CRAF$^{R89L}$in cells with or without 885-A | $F_{(1.8)} =$ 28.9491 interaction | 0.7835 | 1.9023 | 90.2% | 7 (4 groups) |

- Bonferroni- corrected planned comparisons; alpha error = 0.025

| Group 1 | Group 2 | Effect size d | A priori power | Sample size per group |
|---|---|---|---|---|
| CRAF +885A | CRAF +DMSO | 9.2367 | 88.1% | 2 |
| R89L +885A | R89L +DMSO | 4.9941 | 96.0% | 3 |

- 28% variance:
  - ANOVA: Fixed effects, special, main effects, and interactions; alpha error = 0.05

| Groups | F test statistic | Partial eta$^2$ | Effect size $f$ | A priori power | Total sample size |
|---|---|---|---|---|---|
| myc-tagged CRAF$^{WT}$ or CRAF$^{R89L}$in cells with or without SB590885 | $F_{(1.8)} =$8.311 interaction | .05094 | 1.0191 | 82.5% | 11 (4 groups) |

- Bonferroni- corrected planned comparisons; alpha error = 0.025

| Group 1 | Group 2 | Effect size d | A priori power | Sample size per group |
|---|---|---|---|---|
| CRAF +885A | CRAF +DMSO | 4.9482 | 95.8% | 3 |
| R89L +885A | R89L +DMSO | 2.6754 | 90.1% | 5 |

- 40% variance:
  - ANOVA: Fixed effects, special, main effects, and interactions; alpha error = 0.05

| Groups | F test statistic | Partial eta$^2$ | Effect size f | A priori power | Total sample size |
|---|---|---|---|---|---|
| myc-tagged CRAF$^{WT}$ or CRAF$^{R89L}$ in cells with or without SB590885 | $F_{(1,8)} = 4.071$ interaction | 0.3372 | 0.7133 | 80.3% | 18 (4 groups) |

- Bonferroni- corrected planned comparisons; alpha error = 0.025

| Group 1 | Group 2 | Effect size d | A priori power | Sample size per group |
|---|---|---|---|---|
| CRAF +885A | CRAF +DMSO | 3.4638 | 94.0% | 4 |
| R89L +885A | R89L +DMSO | 1.8728 | 81.2% | 7 |

- Based on these power calculations, we will run the experiment three times. Each time, we will quantify band intensity. We will determine the standard deviation of band intensity across the biological replicates and combine this with the effect size from the original study to calculate the number of replicates necessary to reach a power of at least 80%. We will then perform additional replicates, if required, to ensure the experiment has more than 80% power to detect the original effect.

## Protocol 4
### Summary of original data

- The original data presented is qualitative (images of Western blots). We used Image Studio Lite (LICOR) to perform densitometric analysis of the presented bands. We then performed a priori power calculations with a range of assumed standard deviations to determine the number of replicates to perform.
- Note: band intensity quantified from the image reported in Figure 3B:

| Target | Myc-eptitope tagged vector | Drug | Band intensity normalized to IP myc | Assumed N |
|---|---|---|---|---|
| CRAF | BRAF | 885-A | 0.0320 | 3 |
| | | DMSO | 0.6015 | 3 |
| | R188L | 885-A | 0.0164 | 3 |
| | | DMSO | 0.1012 | 3 |

### Test family

- Two-way ANOVA (2 x 2) on CRAF values followed by Bonferroni corrected comparisons for the following groups:
  - Compare the band intensity of BRAF in myc-tagged BRAF$^{WT}$ or BRAF$^{R188L}$ in cells treated with or without 885-A

### Power calculations

- Power calculations were performed using R software version 3.1.2 (*R Core Team, 2014*) and G*Power (version 3.1.7) (*Faul et al., 2007*)
- 2% variance:

- <u>ANOVA: Fixed effects, special, main effects, and interactions; alpha error = 0.05</u>

| Groups | F test statistic | Partial eta$^2$ | Effect size $f$ | A priori power | Total sample size |
|---|---|---|---|---|---|
| myc-tagged BRAF$^{WT}$ or BRAF$^{R188L}$ in cells with or without 885-A | $F_{(1.8)}$ = 4718.4 (interaction) | 0.998 | 24.28 | 99.9% | 5 |

- Bonferroni- corrected planned comparisons; alpha error = 0.025

| Group 1 | Group 2 | Effect size d | A priori power | Sample size per group |
|---|---|---|---|---|
| BRAF +885A | BRAF +DMSO | 66.85 | 99.9% | 2 |
| R188L +885A | R188L +DMSO | 58.51 | 99.9% | 2 |

- <u>15% variance:</u>
  - <u>ANOVA: Fixed effects, special, main effects, and interactions; alpha error = 0.05</u>

| Groups | F test statistic | Partial eta$^2$ | Effect size $f$ | A priori power | Total sample size |
|---|---|---|---|---|---|
| myc-tagged BRAF$^{WT}$ or BRAF$^{R188L}$ in cells with or without 885-A | $F_{(1.8)}$ = 83.88 interaction | 0.913 | 3.238 | 95.6% | 6 |

- Bonferroni- corrected planned comparisons; alpha error = 0.025

| Group 1 | Group 2 | Effect size d | A priori power | Sample size per group |
|---|---|---|---|---|
| BRAF +885A | BRAF +DMSO | 8.914 | 86.3% | 2 |
| R188L +885A | R188L +DMSO | 7.801 | 99.9% | 3 |

- <u>28% variance:</u>
  - <u>ANOVA: Fixed effects, special, main effects, and interactions; alpha error = 0.05</u>

| Groups | F test statistic | Partial eta$^2$ | Effect size $f$ | A priori power | Total sample size |
|---|---|---|---|---|---|
| myc-tagged BRAF$^{WT}$ or BRAF$^{R188L}$ in cells with or without 885-A | $F_{(1.8)}$ = 24.07 interaction | 0.750 | 1.734 | 85.0% | 7 |

- Bonferroni- corrected planned comparisons; alpha error = 0.025

| Group 1 | Group 2 | Effect size d | A priori power | Sample size per group |
|---|---|---|---|---|
| BRAF +885A | BRAF +DMSO | 4.775 | 94.5% | 3 |
| R188L +885A | R188L +DMSO | 4.179 | 87.8% | 3 |

- <u>40% variance:</u>
  - <u>ANOVA: Fixed effects, special, main effects, and interactions; alpha error = 0.05</u>

| Groups | F test statistic | Partial eta² | Effect size f | A priori power | Total sample size |
|---|---|---|---|---|---|
| myc-tagged BRAF^WT or BRAF^R188L in cells with or without 885-A | $F_{(1.8)}$ = 11.79 interaction | 0.596 | 1.214 | 82.7% | 9 |

- Bonferroni- corrected planned comparisons; alpha error = 0.025

| Group 1 | Group 2 | Effect size d | A priori power | Sample size per group |
|---|---|---|---|---|
| BRAF +885A | BRAF +DMSO | 3.343 | 92.3% | 4 |
| R188L +885A | R188L +DMSO | 2.925 | 83.9% | 4 |

- Based on these power calculations, we will run the experiment three times. Each time, we will quantify band intensity. We will determine the standard deviation of band intensity across the biological replicates and combine this with the effect size from the original study to calculate the number of replicates necessary to reach a power of at least 80%. We will then perform additional replicates, if required, to ensure the experiment has more than 80% power to detect the original effect.

## Protocol 5
### Summary of original data

- The original data presented is qualitative (images of Western blots). We used Image Studio Lite (LICOR) to perform densitometric analysis of the presented bands. We then performed a priori power calculations with a range of assumed standard deviations to determine the number of replicates to perform.
- Note: band intensity quantified from the image reported in Figure 4D
  - The band intensities for two groups were beyond the dynamic range for intensity calculation:
    - IP myc-tagged BRAF in cells transfected with the BRAF mutant (D594A): In this case, we used the value for band intensity of IP myc-tagged BRAF in cells transfected with wild type BRAF as an estimate. Since the band for wild type BRAF transfected cells was less intense, this underestimates the effect size, so we are likely overestimating the sample size required.

| Target | Vector | Band intensity normalized to IP myc | Assumed N |
|---|---|---|---|
| IP CRAF | BRAF | 0.164 | 3 |
| | D594A | 0.739 | 3 |

### Test family

- Unpaired two-tailed Welch's t-test, alpha error = 0.05.

### Power calculations

- Power calculations were performed using R software version 3.2.2 (*R Core Team, 2014*)

| Group 1 | Group 2 | Variance estimate | Effect size (Glass' $\Delta$)[1] | A priori power | Sample size per group |
|---------|---------|-------------------|----------------------------------|----------------|------------------------|
| BRAF^WT | BRAF^D594A | 2% | 175.30 | >99.9% | 2 |
| | | 15% | 23.374 | 89.9% | 2 |
| | | 28% | 12.522 | 93.3% | 3 |
| | | 40% | 8.7652 | 90.8% | 4 |

[1] The BRAF group SD was used as the divisor.

- Based on these power calculations, we will run the experiment three times. Each time, we will quantify band intensity. We will determine the standard deviation of band intensity across the biological replicates and combine this with the effect size from the original study to calculate the number of replicates necessary to reach a power of at least 80%. We will then perform additional replicates, if required, to ensure the experiment has more than 80% power to detect the original effect.

## Acknowledgements

The Reproducibility Project: Cancer Biology core team would like to thank the original authors, in particular Sonja Heidorn and Richard Marais, for generously sharing critical information as well as reagents to ensure the fidelity and quality of this replication attempt. We thank Courtney Soderberg at the Center for Open Science for assistance with statistical analyses. We also thank the following companies for generously d9onating reagents to the Reproducibility Project: Cancer Biology; American Type and Tissue Collection (ATCC), Applied Biological Materials, BioLegend, Charles River Laboratories, Corning Incorporated, DDC Medical, EMD Millipore, Harlan Laboratories, LI-COR Biosciences, Mirus Bio, Novus Biologicals, Sigma-Aldrich, and System Biosciences (SBI).

## Additional information

### Group author details

Reproducibility Project: Cancer Biology

Elizabeth Iorns: Science Exchange, Palo Alto, United States; William Gunn: Mendeley, London, United Kingdom; Fraser Tan: Science Exchange, Palo Alto, United States; Joelle Lomax: Science Exchange, Palo Alto, United States; Stephen R Williams: Science Exchange, Palo Alto, United States; Nicole Perfito: Science Exchange, Palo Alto, United States; Timothy Errington: Center for Open Science, Charlottesville, United States

### Competing interests

AB, MA: Shakti BioResearch LLC, is a Science Exchange lab. RP:CB: EI, FT, JL, and NP are employed by and hold shares in Science Exchange Inc. RP:CB employed by and holds shares in Science Exchange Inc The other authors declare that no competing interests exist.

### Funding

| Funder | Author |
|--------|--------|
| Laura and John Arnold Foundation | Nicole Perfito |

The funders had no role in study design, data collection and interpretation, or the decision to submit the work for publication.

## Author contributions

AB, MA, HM, Drafting or revising the article; RP:CB, NP, Conception and design, Drafting or revising the article

---

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
