## [Decision Letter]

Thank you for submitting your work entitled "Registered report: Kinase-Dead BRAF and Oncogenic RAS Cooperate to Drive Tumor Progression through CRAF" for consideration by *eLife*. Your article has been favorably evaluated by Charles Sawyers (Senior editor) and four reviewers, one of whom is a member of our Board of Reviewing Editors.

The reviewers have discussed the reviews with one another and the Reviewing editor has drafted this decision to help you prepare a revised submission.

The 2010 paper, "Kinase-Dead BRAF and Oncogenic RAS Cooperate to Drive Tumor Progression through CRAF" by Heidorn et al., showed 1) selective BRAF inhibitors in the presence of oncogenic RAS led to RAS-dependent BRAF:CRAF dimerization and RAS-dependent activation of MEK-ERK signaling; 2) kinase-dead BRAF mimics the effects of BRAF-selective inhibitors; and 3) kinase-dead BRAF and oncogenic RAS cooperate to induce melanoma in vivo. The authors of the Reproducibility Project propose to replicate 4 figures. The first three figures (Figures 1A, 1B, and 3A) will support the first finding of the paper and the last figure of the proposal (Figure 4D) will support the second finding. While the proposed replicated figures do not address the third conclusion at all, it is understood that duplication of animal experiments might be considered to be unethical.

Figure 1A demonstrates that selective BRAF inhibitors deplete ERK signaling in BRAF mutant cell lines while promoting signaling in cells with WT BRAF and mutant RAS (NRAS). The authors propose to use only two lines, A375 which is BRAF^V600E^ and D04 which is Ras mutant. They will only use PD, Sorafenib and SB and have chosen to not use PLX4720. The authors might wish to reconsider their decision to not repeat with PLX4720. The data have been repeated many times by others, but the data, that PLX induces paradoxical activation and this is dependent on CRAF is a key finding of this figure.

Figure 1B shows that activation of MEK-ERK signaling was abrogated in the mutant RAS cell line (D04) by transiently depleting NRAS before treatment with the BRAF inhibitor (885-A).

Figure 3A confirms that CRAF must interact with RAS to promote BRAF:CRAF dimerization. Finally, Figure 4D shows that kinase-dead BRAF (BRAF^D594A^), but not BRAF^WT^, mimics BRAF inhibition and heterodimerizes with CRAF in NRAS-mutant cells (D04).

Figure 4D: Here it is shown that a specific kinase dead form of BRAF^D594A^, can bind constitutively to CRAF. This is a straightforward experiment and suggests that BRAF inhibition is sufficient to stimulate dimer formation with CRAF.

The reviewers recommend that the replication study should be expanded to include:

Figure 2A: The authors show that Sorafenib strongly induces dimers between BRAF and CRAF. This was confirmed by Rosen but Therrien suggests that it doesn't induce strong dimers. Thus, it would be of interest to validate this finding.

Figure 2B: The authors suggest that the inability to detect PLX induced dimers between BRAF and CRAF is feedback phosphorylation because of pathway activation. thus, they show that MEK inhibition (which blocks downstream activation), allows for weak detection of PLX induced BRAF/CRAF dimers. This explained how PLX could induce paradoxical activation. However, recent structural studies and work from the Theirrien group suggests that PLX prevents dimer formation because it moves the aC helix. This model is in conflict with the data in Figure 2B. The possibility that weak dimers are formed which are inhibited by MEK activation could be a simple resolution to this issue.

Figure 3B: A key finding of the original study is that RAS interaction with both CRAF and BRAF is required to induce BRAF:CRAF dimerization in the presence of a BRAF inhibitor.

Based on the reasoning outlined above, the reviewers recommend that the replication study should be expanded to include Figures 2A, 2B & 3B.

Specific comments on detailed protocols:

1) Protocol 2:

A) In Step 2, the authors should use a non-targeting siRNA in addition to their "Mock Transfection" control. It is unclear why the authors use the term "Mock siRNA" in their confirmatory analysis plan when their mock transfection clearly states 0.6 μL of media (not non-targeting siRNA).

2) Protocol 3:

A) There is a minor concern that a different transfection protocol will be used. Nucleofection will be replaced with a lipid based transfection reagent. Significant differences in expression could lead to differences in results.

B) It is unclear why the authors list an NRAS antibody in Protocol 3 and not a CRAF antibody when the intent to the protocol is to immunoprecipitate CRAF.

C) Step 3b states: "Freeze the remaining lysate (-20C) to be used for Step 3. Save an aliquot of lysate to run as a control in Step 4b." Are the authors referring to Step 4 in the first sentence? Are the authors proposing to freeze the lysate before an IP? There is substantial concern that protein-protein interactions will not survive the freeze/thaw of the lysate.

D) Step 4a, why are the authors using both anti-CRAF (C-20) and anti-myc in the same IP?

3) Protocol 4:

A) The kinase-dead BRAF mutant is listed as "VRAF^D594A^" in the Materials and Reagents table.

B) The authors are proposing to freeze the lysate (Step2) before performing the IP (Step 3). As in point 2b above, there is substantial concern that protein-protein interactions will not survive the freeze/thaw of the lysate.

Statistical Comments:

For protocol 1 & 3, the authors propose use ANOVA to analyze the data. Please check for outliers and make sure that the data do not violate the assumptions of the ANOVA: normality and homoscedasticity. If the data do not fit the assumptions well enough, try to find a data transformation that makes them fit. If this doesn't work, then you will need to apply a nonparametric counterpart of ANOVA.

For protocol 2, the authors propose use MANOVA to analyze the data.

In addition to what mentioned above, MANOVA assumes that covariances of dependent variables are homogeneous across the cells of the design and that the dependent variables should not be too correlated to each other. Furthermore, it assumes that there are linear relationships among all pairs of dependent variables. Please verify these assumptions before applying MANOVA.

For protocol 4, the authors propose use unpaired student t-test to analyze the data. We would suggest the authors to use either unequal variance welch t-test or use a test for equal variances followed by appropriate test depending on the outcome of the equal variance test. Please adjust power calculation for protocol 4 accordingly.

---

## [Author Response]

The 2010 paper, "Kinase-Dead BRAF and Oncogenic RAS Cooperate to Drive Tumor Progression through CRAF" by Heidorn et al., showed 1) selective BRAF inhibitors in the presence of oncogenic RAS led to RAS-dependent BRAF:CRAF dimerization and RAS-dependent activation of MEK-ERK signaling; 2) kinase-dead BRAF mimics the effects of BRAF-selective inhibitors; and 3) kinase-dead BRAF and oncogenic RAS cooperate to induce melanoma in vivo. The authors of the Reproducibility Project propose to replicate 4 figures. The first three figures (Figures 1A, 1B, and 3A) will support the first finding of the paper and the last figure of the proposal (Figure 4D) will support the second finding. While the proposed replicated figures do not address the third conclusion at all, it is understood that duplication of animal experiments might be considered to be unethical.Figure 1A demonstrates that selective BRAF inhibitors deplete ERK signaling in BRAF mutant cell lines while promoting signaling in cells with WT BRAF and mutant RAS (NRAS). The authors propose to use only two lines, A375 which is BRAF V600E and D04 which is Ras mutant. They will only use PD, Sorafenib and SB and have chosen to not use PLX4720. The authors might wish to reconsider their decision to not repeat with PLX4720. The data have been repeated many times by others, but the data, that PLX induces paradoxical activation and this is dependent on CRAF is a key finding of this figure.

We agree that the key finding to this figure is demonstrating the paradoxical activation of MEK/ERK signaling in cells with WT BRAF and mutant RAS. The authors tested two specific BRAF inhibitors (PLX4720 and 885-A) and compared their effects to a compound that represses both BRAF and CRAF (Sorafenib) in cells with (A375) or without (D04) BRAF mutation to demonstrate that cells with active RAS (D04) respond with increased MEK/ERK signaling to *specific* BRAF inhibitors, but not with a pan RAF inhibitor. Additionally, 885-A is utilized in further experiments included in this replication attempt, specifically Figure 1B, 3A, and 3B. We agree that including all of the compounds tested would be of general interest and limits the scope of what can be analyzed by the project, but we are attempting to identify a balance of breadth of sampling for general inference with sensible investment of resources on replication projects.

Figure 1B shows that activation of MEK-ERK signaling was abrogated in the mutant RAS cell line (D04) by transiently depleting NRAS before treatment with the BRAF inhibitor (885-A). Figure 3A confirms that CRAF must interact with RAS to promote BRAF:CRAF dimerization. Finally, Figure 4D shows that kinase-dead BRAF (BRAFD594A), but not BRAFWT, mimics BRAF inhibition and heterodimerizes with CRAF in NRAS-mutant cells (D04). Figure 4D: Here it is shown that a specific kinase dead form of BRAF, D594A, can bind constitutively to CRAF. This is a straightforward experiment and suggests that BRAF inhibition is sufficient to stimulate dimer formation with CRAF. The reviewers recommend that the replication study should be expanded to include:Figure 2A: The authors show that Sorafenib strongly induces dimers between BRAF and CRAF. This was confirmed by Rosen but Therrien suggests that it doesn't induce strong dimers. Thus, it would be of interest to validate this finding.Figure 2B: The authors suggest that the inability to detect PLX induced dimers between BRAF and CRAF is feedback phosphorylation because of pathway activation. thus, they show that MEK inhibition (which blocks downstream activation), allows for weak detection of PLX induced BRAF/CRAF dimers. This explained how PLX could induce paradoxical activation. However, recent structural studies and work from the Theirrien group suggests that PLX prevents dimer formation because it moves the aC helix. This model is in conflict with the data in Figure 2B. The possibility that weak dimers are formed which are inhibited by MEK activation could be a simple resolution to this issue.Figure 3B: A key finding of the original study is that RAS interaction with both CRAF and BRAF is required to induce BRAF:CRAF dimerization in the presence of a BRAF inhibitor. Based on the reasoning outlined above, the reviewers recommend that the replication study should be expanded to include Figures 2A, 2B & 3B.

We appreciate the comments provided by the reviewers about expanding the experimental work for this replication. We agree that all of the experiments included in the original study are important, and choosing which experiments to replicate has been one of the great challenges of this project. The Reproducibility Project: Cancer Biology (RP:CB) aims to replicate experiments that are impactful, but does not necessarily aim to replicate all the impactful experiments in any given paper. We agree that the exclusion of certain experiments limits the scope of what can be analyzed by the project, but we are attempting to identify a balance of breadth of sampling for general inference with sensible investment of resources on replication projects to determine to what extent the included experiments are reproducible. We agree that one of the key findings of the original paper was that both BRAF and CRAF must bind to RAS to create the proposed stable complex, so have included this additional experiment, reported in Figure 3B of the original study, into the revised Registered Report. However, we did not include Figures 2A and 2B, since they are not as central to the main findings of the original study even though they would be of interest to replicate considering new evidence reported by other groups. As such, we will restrict our analysis to the experiments being replicated and will not include discussion of experiments not being replicated in this study.

Specific comments on detailed protocols: 1) Protocol 2:

*A) In Step 2, the authors should use a non-targeting siRNA in addition to their "Mock Transfection" control. It is unclear why the authors use the term "Mock siRNA" in their confirmatory analysis plan when their mock transfection clearly states 0.6 μL of media (not non-targeting siRNA).*

The reviewers make a good point. We have removed the mock transfection and replaced the scrambled siRNA as the relevant control. We have replaced “Mock” with “Control” in the text.

2) Protocol 3:

*A) There is a minor concern that a different transfection protocol will be used. Nucleofection will be replaced with a lipid based transfection reagent. Significant differences in expression could lead to differences in results.*

We have expanded upon this potential impact on the outcome in the known differences section of the protocol. Since the original expression levels of are unknown, such as expression above endogenous, even if the same transfection protocol was to be used, differences in expression of the original compared to the replication could occur.

*B) It is unclear why the authors list an NRAS antibody in Protocol 3 and not a CRAF antibody when the intent to the protocol is to immunoprecipitate CRAF.*

Thank you for catching this error. NRAS should not be included in the Reagents list. We have removed it in the revised manuscript. The original authors only probed for myc-tagged CRAF and endogenous BRAF.

C) Step 3b states: "Freeze the remaining lysate (-20C) to be used for Step 3. Save an aliquot of lysate to run as a control in Step 4b." Are the authors referring to Step 4 in the first sentence? Are the authors proposing to freeze the lysate before an IP? There is substantial concern that protein-protein interactions will not survive the freeze/thaw of the lysate.

We agree this section is confusing. We’ve removed the reference to freezing the sample and reworded this section to state the procedure more clearly.

D) Step 4a: why are the authors using both anti-CRAF (C-20) and anti-myc in the same IP?

The incubation should only use α-myc antibody. The reference to anti-CRAF has been removed.

3) Protocol 4:

*A) The kinase-dead BRAF mutant is listed as "VRAF^D594A^" in the Materials and Reagents table.*

We have corrected this typo.

B) The authors are proposing to freeze the lysate (Step2) before performing the IP (Step 3). As in point 2b above, there is substantial concern that protein-protein interactions will not survive the freeze/thaw of the lysate.

We agree and have removed the freezing step.

Statistical Comments: For protocol 1 & 3, authors propose use ANOVA to analyze the data. Please check for outliers and make sure that the data do not violate the assumptions of the anova: normality and homoscedasticity. If the data do not fit the assumptions well enough, try to find a data transformation that makes them fit. If this doesn't work, then you will need to apply a nonparametric counterpart of ANOVA.

We agree and at the time of analysis, we will assess the normality and homoscedasticity of the data. If necessary, we will perform the appropriate transformation in order to proceed with the proposed statistical analysis or apply a nonparametric counterpart of ANOVA We will note any changes or transformations made. We have also updated the manuscript to address this point.

For protocol 2, authors propose use MANOVA to analyze the data.

In addition to what mentioned above, MANOVA assumes that covariances of dependent variables are homogeneous across the cells of the design and that the dependent variables should not be too correlated to each other. Furthermore, it assumes that there are linear relationships among all pairs of dependent variables. Please verify these assumptions before applying MANOVA.

We agree and at the time of analyze we will check the additional assumptions of a MANOVA. We have updated the manuscript to address this point.

For protocol 4, the authors propose use unpaired student t-test to analyze the data. We would suggest the authors to use either unequal variance welch t-test or use a test for equal variances followed by appropriate test depending on the outcome of the equal variance test. Please adjust power calculation for protocol 4 accordingly.

Thank you for this suggestion. We have updated the manuscript and power calculations to reflect a Welch’s t-test.